

# Uncertainties in projections of the Baltic Sea ecosystem driven by an ensemble of global climate models

Sofia Saraiva[1,3], H.E. Markus Meier[2,3], Helén Andersson[3], Anders Höglund[3], Christian Dieterich[3], Robinson Hordoir[3], and Kari Eilola[3]

[1]University of Lisbon, Instituto Superior Técnico, Environment and Energy Section, Av. Rovisco Pais, 1,1049-001 Lisbon, Portugal.
[2]Department of Physical Oceanography and Instrumentation, , Leibniz Institute for Baltic Sea Research Warnemünde, , 18119 Rostock, Germany.
[3]Swedish Meteorological and Hydrological Institute , 60176 Norrköping, Sweden.

*Correspondence to:* Sofia Saraiva (sofia.maretec@tecnico.ulisboa.pt)

**Abstract.** Many coastal seas worldwide are affected by human impacts such as eutrophication, causing, inter alia, oxygen depletion and extensive areas of hypoxia. Depending on the region, global warming may reinforce these environmental changes by reducing air-sea oxygen fluxes, intensifying internal nutrient cycling and increasing river-borne nutrient loads. The development of appropriate management plans to more effectively protect the marine environment requires projections of future marine ecosystem states. However, projections with regional climate models commonly suffer from shortcomings in the driving global General Circulation Models (GCMs). The differing sensitivities of GCMs to increased greenhouse gas emissions impact regional projections considerably. In this study, we focused on one of the most threatened coastal seas, the Baltic Sea, and estimated uncertainties in projections due to GCM deficiencies relative to uncertainties caused by future greenhouse gas emissions and nutrient load scenarios. To address the latter, transient simulations of the period 1975-2098 were performed using the initial conditions from an earlier reconstruction with the same Baltic Sea model (starting in 1850). To estimate the impacts of GCM deficiencies, dynamical downscaling experiments with four driving global models were carried out for two greenhouse gas emission scenarios, RCP 4.5 and 8.5, and for three nutrient load scenarios covering the plausible range between low and high loads. The results of primary production, nitrogen fixation, and hypoxic areas show that uncertainties caused by the various nutrient load scenarios are greater than the uncertainties due to global model deficiencies and future greenhouse gas emissions. In all scenario simulations, a proposed nutrient load abatement strategy, i.e., the Baltic Sea Action Plan, will lead to a significant improvement in the overall environmental state. However, the projections cannot provide detailed information on the timing and the reductions of future hypoxic areas due to uncertainties in salinity projections caused by uncertainties in projections of the regional water cycle and of the global mean sea level rise.

*Copyright statement.* TEXT



# 1 Introduction

Regional projections of future climate based on dynamical downscaling of global model results using regional climate models suffer from considerable uncertainties caused by (1) shortcomings of global and regional climate models, (2) unknown future greenhouse gas emissions, (3) natural variability and (4) experimental design (e.g. Déqué et al., 2007; Kjellström et al., 2011;

Mathis et al., 2017). Uncertainties in regional atmospheric projections have been studied, e.g., by (Christensen et al., 2010; Jacob et al., 2014). Global climate models are based on General Circulation Models (GCMs) or even Earth System Models (ESMs) and are useful tools to address climate variability on the global scale. However, they have still significant shortcomings on regional scales, inter alia, because their horizontal grids are too coarse to resolve details of the orography and the land-sea mask in the Baltic Sea region, which are important to the regional climate (Stocker et al., 2013) and oceanographic conditions.

Further, as the socioeconomic development in the catchment area of the coastal seas is unknown, future nutrient loads from land and atmospheric depositions of nitrogen and phosphorus are also unknown, contributing to the uncertainties of the projections of the marine ecosystems (e.g. Meier et al., 2011c). The Baltic Sea is a semi-enclosed coastal sea with a large catchment area located in northern Europe (Sjöberg, 2016). The Baltic Sea region is divided into two sub-regions. Extensive forests, low population density, mostly rocky coasts and subarctic winter climate characterize the north. On the other hand, the south is

characterized by agricultural land, high population density, mostly sandy coasts and a moderate winter climate. Approximately 90 million people live in the catchment area of the Baltic, creating a considerable impact on the marine environment (H. and Öhman, 2014). Reinforced river-borne nutrient loads from agriculture and sewage treatment plants since the 1940/50s caused the world largest anthropogenic-induced hypoxic bottoms (Conley et al., 2009; Gustafsson et al., 2012; Meier et al., 2012a; Conley, 2012; Carstensen et al., 2014). In addition to environmental pressures, the Baltic Sea is affected by global warming

more than other coastal seas, perhaps because of its hydrodynamic features and land-locked location (The BACC II Author Team, 2015). During 1982-2006, the Baltic Sea warmed the most of the large marine ecosystems investigated by Belkin (2009). By approximately 18 years ago, future projections of the Baltic Sea were being carried out with the specific aim of supporting the design of appropriate management plans to more effectively protect the marine environment. First, projections were made with pure hydrodynamical models (Omstedt et al., 2000; Meier, 2002; Ralf Döscher, 2004; H.E. Markus Meier, 2004; Meier,

2006) and, later, with coupled physical-biogeochemical models (Neumann, 2010; Meier et al., 2011c, a). The first scenario simulations were based on only one GCM and one greenhouse gas emission scenario, covering only 10-year time slices (Omstedt et al., 2000; Meier, 2002). Meier (2006) and Meier et al. (2011c) assumed that the variabilities on time scales longer than one year do not change. Hence, uncertainties, e.g., those due to natural variability, could not be appropriately addressed. To overcome this shortcoming, more recent approaches based on transient simulations from the present to future climates utilizing

mini-ensembles consisting of two GCMs and two emission scenarios (e.g. Neumann, 2010; Friedland et al., 2012; Meier et al., 2012b; Ryabchenko et al., 2016) were used; in addition, three Baltic Sea ecosystem models were implemented (e.g. Meier et al., 2011a, 2012a, c). Omstedt et al. (2012) even used three GCMs and three emission scenarios, although they applied a regional atmospheric model instead of a regional coupled atmosphere-ocean model for the dynamical downscaling. By using a regional atmosphere model, the projected sea surface temperatures (SST) were based on the SSTs from GCMs that do not take



the regional details of the Baltic Sea region into account, causing considerable deficiencies in projections (e.g. Meier et al., 2012c).

Hence, uncertainties of Baltic Sea projections originating from GCM deficiencies have not been thoroughly assessed. An exception is the study by Meier et al. (2006), who studied the spread of a multi-model ensemble consisting of 16 scenario sim-

ulations based on seven regional models, five global models and two emission scenarios. However, Meier et al. (2006) focused only on salinity and neglected changes in variability on time scales longer than one year by applying the delta approach. In addition, changing biogeochemical cycles were not addressed. In all other dynamical downscaling studies of the Baltic Sea, only a limited number of GCMs was used and only GCMs that showed high quality atmospheric variables over the Baltic Sea were selected (e.g. Meier et al., 2011b). More advanced methods that use hierarchical clustering to select an optimum subset

from the entire ensemble to estimate uncertainties by a minimum number of scenario simulations (Wilcke and Bärring, 2016) have not yet been applied to the Baltic Sea.

In this study, we focus on uncertainties in Baltic Sea water quality projections. Compared to the earlier studies summarized above, the new features of this study are:

1. transient simulations for the period 1850-2098 including a spin-up with reconstructed forcing 1850-1975;

2. consistent simulations without bias correction except for the wind speed;

3. dynamical downscaling of four GCMs with the aim of estimating uncertainties;

4. revised, more plausible nutrient load scenarios taking the latest load observations into account;

5. two greenhouse gas emission scenarios corresponding to the representative concentration pathways (RCPs) 4.5 and 8.5
(Moss et al., 2010);

6. improved versions of the coupled physical-biogeochemical model of the Baltic Sea with the aim of reducing model shortcomings;

7. improved versions of the global models from the Coupled Model Intercomparison Project 5 (CMIP5) of the Intergovernmental Panel of Climate Change (IPCC) (Stocker et al., 2013).

The paper is organized as follows. In Section 2, the regional climate ocean model, the regional coupled atmosphere-ocean model, driving GCMs, greenhouse gas emission and nutrient load scenarios and the experimental setup are introduced. In Section 3, the results of future projections for temperature, salinity, selected biogeochemical fluxes (primary production and nitrogen fixation) and hypoxic areas are presented. In Section 4, the suspected shortcomings of the study are discussed. In Section 5, some conclusions of the study are drawn.



## 2 Methods

### 2.1 Baltic Sea Model

In this study, a three-dimensional ocean circulation model is used in climate simulations for the period 1975-2098. RCO-SCOBI consists of the physical Rossby Center Ocean (RCO) (Meier et al., 2003; Meier, 2007) and the Swedish Coastal and

Ocean Biogeochemical (SCOBI) models (Eilola et al., 2009; Almroth-Rosell et al., 2011). The model domain covers the Baltic Sea area with an open boundary in northern Kattegat (Fig.1). The horizontal and vertical resolutions are 3.7 km and 3 m (corresponding to 83 depth levels), respectively. In the water column, the biogeochemical model SCOBI describes the dynamics of nitrate, ammonium, phosphate, three phytoplankton groups (diatoms, flagellates and others, and cyanobacteria), zooplankton, detritus, oxygen and hydrogen sulfide as negative oxygen equivalents ($1mLH_2SL^{-1} = -2mLO_2L^{-1}$). In the present version,

the nitrogen and phosphorus detritus were separated according to (Savchuk, 2002). The sediment contains nutrients in the form of benthic nitrogen and benthic phosphorus. With the help of a simplified wave model, the resuspension of organic matter is calculated (Almroth-Rosell et al., 2011). RCO-SCOBI has previously been evaluated and applied in numerous long-term climate studies, e.g., Meier et al. (2003); Meier (2007); Meier et al. (2011a, 2012b); Eilola et al. (2009); Eilola et al. (2011); Almroth-Rosell et al. (2011); Schimanke and Meier (2016).

### 2.2 Regional climate data sets

The Baltic Sea model was forced by (1) atmospheric surface fields from a regional coupled atmosphere-ocean climate model (RCM) driven by lateral boundary conditions from GCMs and by (2) runoff and (3) nutrient loads from a regional hydrological model also forced by regionalized atmospheric data from the same GCMs. The RCM is RCA4-NEMO applied to

the EURO-CORDEX domain (Jacob et al., 2014) with an interactively coupled Baltic Sea and North Sea (Dieterich et al., 2006; Wang et al., 2015; Gröger et al., 2015). The regional atmosphere model RCA4 has a 0.22-degree spherical rotated latitude/longitude grid with 40 vertical levels. The hydrological model is E-HYPE (Hydrological Predictions for the Environment, http://hypeweb.smhi.se), a process-based multi-basin model applied for Europe (Hundecha et al., 2016; Donnelly et al., 2013, 2017). The runoff from each river was corrected (in the historical and future periods) by a factor that corresponds to the ratio

between the total annual flow to the Baltic computed based on observations and on E-HYPE results. This factor was computed from the period 1971-2005, which is the maximum time interval with overlaps between observations and E-HYPE results. This approach has been previously applied for regional climate simulations of the Baltic Sea (e.g. Meier et al., 2012b). Here, improved versions of the regional and global climate models and the results of scenario simulations from the latest IPCC assessment report were used (Stocker et al., 2013). We focused on the greenhouse gas emission scenarios RCP 4.5 and 8.5 (Moss

et al., 2010; Detlef P. et al., 2011; Stocker et al., 2013). RCP 4.5 and 8.5 are medium and high-end scenarios, respectively.

In this study, regionalized atmospheric forcing data for the historical (1976-2005) and future (2006-2098) periods from four GCMs were used: MPI-ESM-LR (https://www.mpimet.mpg.de), EC-EARTH (https://www.knmi.nl), IPSL-CM5A-MR (http://icmc.ipsl.fr/) and HadGEM2-ES (http://www.metoffice.gov.uk). Henceforth, the four models are called Models A, B,



C and D, respectively. The selection of these GCMs follows the approach presented by Wilcke and Bärring (2016). Their aim was to select an ensemble that provides realistic solutions for the regional climate systems of the North Sea and Baltic Sea regions that also reproduce the uncertainties inherent in an ensemble of coupled RCM solutions with just a few ensemble members. The four driving GCMs of this study were selected from the distinct clusters identified by Wilcke and Bärring (2016). The necessary lateral boundary data for RCA4-NEMO from each of these GCMs were provided by the Rossby Center of the Swedish Meteorological and Hydrological Institute (SMHI). The data from the selected GCMs allow the production of scenario simulations with a realistic climate for the North Sea and Baltic Sea (Dieterich et al., submitted manuscript). The RCA4-NEMO results indicate that strong winds are underestimated compared to the high quality reanalysis dataset EURO4M (http://www.euro4m.eu/(. This pattern was previously identified in earlier studies, wherein results were compared with the previous reanalysis ERA40 (Meier et al., 2011b). Therefore, a correction was made by multiplying the portion of the wind speed exceeding $10\ ms^{-1}$ by a factor of 1.6 without altering wind direction.

As examples, the results of the seasonal cycles of regionalized air temperature over the central Baltic and the total runoff in the present and future climates are shown in Figs. 2 and 3, respectively. In the future climate (2069-2098), air temperatures over the central Baltic will increase more in winter than in summer, and runoff from the entire catchment area will increase during winter but decrease during summer. In terms of the annual mean, both temperature and runoff will increase in the future compared to those of the historical climate. During the historical period, annual and monthly biases of both variables were within the range of variability of the observations, i.e., within the range of plus or minus one standard deviation from the monthly mean.

## 2.3 Nutrient load scenarios

Climate projections for the Baltic Sea are carried out under the three nutrient load scenarios described below, spanning a range of plausible future socio-economic conditions. During the historical period (1976-2005), the observed nutrient loads from the Baltic Environmental Database (BED) are used (http://nest.su.se/bed/).

1. Baltic Sea Action Plan (BSAP) (HELCOM, 2013). In this scenario, nutrient loads from rivers and atmospheric deposition in different sub-basins will linearly decrease after 2012 from the current values (average 2010-2012) as estimated by Svendsen et al. (2015) to the maximum allowable input defined by the BSAP until 2020. After 2020, nutrient loads will remain constant until 2098.

2. Reference. In this scenario, E-HYPE projections for future nutrient loads (2006-2098) under the two different greenhouse gas emission scenarios (RCP 4.5 and 8.5) are used, assuming no socio-economic changes compared to the historical period (1976-2005). Hence, e.g., biogeochemical processes, land use and soil properties in each sub-basin do not change over time. Only the impacts of the changing climate on air temperature and precipitation are considered. Atmospheric deposition is constant in time.

3. Worst Case. In this scenario, future nutrient loads calculated with E-HYPE under the RCP 4.5 and 8.5 scenarios (2006-2098) are multiplied by a factor that summarizes the impact of a worst case socio-economic development on current





nutrient loads. The main assumptions and description of this impact factor can be found in (Zandersen et al., in prep).
Correspondingly, future atmospheric deposition is calculated from current deposition assuming the same worst case
socio-economic development (and impact factor) as well. In all three scenarios, nutrient loads in the Baltic Sea will de-
crease in the future following the historical efforts toward nutrient load reductions starting in the 1980s (Fig.4). However,
in the Worst Case scenario the loads are close to the average observed loads during 1976-2005. Seasonal and long-term
changes in the Reference and Worst Case scenarios follow runoff changes caused by changing climates. Only in BSAP is
it assumed that changing climates do not counteract nutrient load abatement strategies. Hence, the latter is an optimistic
scenario.

## 2.4 Experimental setup

Combinations of future climate scenarios (RCP 4.5 and RCP 8.5; for Model C only RCP 4.5 is available) calculated with
the four GCMs and three socio-economic scenarios (BSAP, Reference and Worst Case) result in an ensemble of 21 scenario
simulations (1). All simulations for the historical period of 1975-2005 driven by the four GCMs start from the same initial
conditions in March 1975, which were obtained from a long-term hindcast simulation starting in 1850. The latter simulation
was driven by reconstructed atmospheric, hydrological, and nutrient loads estimated from the available historical observations
(Meier et al., 2012a).

In addition, two sensitivity experiments for Model A under RCP 8.5 (BSAP and Worst Case) with a 1 m higher sea level during
2006-2098 were performed. In these two experiments the thickness of the uppermost layer was increased by 1 m following
Meier et al. (2017) (i.e., a 4 m surface layer instead of 3 m). Hence, during the start of the sensitivity experiment, the difference
between the depth of the pycnocline and the depth of the sills in the entrance area of the Baltic Sea (Fig.1) did not change
compared to the scenario simulation with unchanged mean sea level.

The impacts of climate and nutrient load changes on the marine ecosystem were quantified by comparing various future sce-
narios (2069-2098) with the historical period of the GCM driven climate simulations (1976-2005) in terms of the annual and
seasonal changes in the physical and biogeochemical variables. We focus our analysis on the changes and uncertainties of water
temperature and salinity as well as on the environmentally important indicators, such as nitrogen fixation, primary production
and hypoxic areas. For the evaluation results of the model simulations during the historical period, the reader is referred to an
accompanying paper (Saraiva et al., submitted manuscript) and to the supplementary material.

To quantify the uncertainties (spread) in the projected changes we follow the approach by Karppanen2006 (2006) and K. Ru-
osteenoja (2016). For the evaluation of uncertainty in the 30-year mean changes between the future (2069-2098) and historical
(1976-2005) climates we calculate and compare the variances of changes caused by each of the different factors; GCMs ($\sigma_1^2$),
RCPs ($\sigma_2^2$), nutrient loads ($\sigma_3^2$), and global mean sea level rise ($\sigma_4^2$).

$$\sigma_1^2: = \frac{1}{\sum_{l=1}^{L}\sum_{k=1}^{K}\sum_{m=1}^{M}(N_{m,k,l}-1)}\sum_{l=1}^{L}\sum_{k=1}^{K}\sum_{m=1}^{M}\sum_{n=1}^{N_{m,k,l}}\left(y_{n,m,k,l}-\bar{y}_{m,k,l}\right)^2$$
$$\sigma_2^2: = \frac{1}{\sum_{n=1}^{N}\sum_{l=1}^{L}\sum_{k=1}^{K}(M_{k,l,n}-1)}\sum_{n=1}^{N}\sum_{l=1}^{L}\sum_{k=1}^{K}\sum_{m=1}^{M_{k,l,n}}\left(y_{m,k,l,n}-\bar{y}_{k,l,n}\right)^2$$

etc.



with the mean value for all four indices n, m, k and l

$$\bar{y}_{m,k,l} := \frac{1}{N_{m,k,l}} \sum_{n=1}^{N_{m,k,l}} y_{n,m,k,l}$$
$$\bar{y}_{k,l,n} := \frac{1}{M_{k,l,n}} \sum_{m=1}^{M_{k,l,n}} y_{n,m,k,l}$$

etc.

$N = 4$, $M = 2$, $K = 3$, and $L = 2$ are the total numbers of global models, greenhouse gas emission scenarios, nutrient load scenarios and sea level rise scenarios, respectively. $M_{k,l,n}$ is the number of global models for the greenhouse gas emission scenario m, the nutrient load scenario $k$ and the sea level rise scenario $l$. For instance, global mean sea level scenarios exist only for model A together with either BSAP or Worst Case. Further, for model C the RCP 8.5 scenario is not available. Hence, the number of terms in the equations above are smaller than the product $N * M * K * L$. $M_{k,l,n}$ is defined correspondingly. In case of $\sigma_1^2$, the variances caused by global models are summed for all combinations of RCPs, nutrient loads and sea level rise experiments. Finally, all variances of changes in primary production, nitrogen fixation and hypoxic area are normalized by the corresponding variance based upon the changes in all 23 simulations.

## 3 Results of Future Projections

### 3.1 Temperature and Salinity

According to our ensemble of scenario simulations, water temperature will increase with time as a direct consequence of the increase in air temperature projected by the GCMs (Fig.5). The ensemble mean of the Baltic Sea volume averaged temperature change (and its standard deviation) between future (2069–2098) and historical (1976–2005) conditions amounts to 1.6 ± 0.5 °C in RCP 4.5 and to 2.7± 0.4 °C in RCP 8.5. The largest changes in SST follow the spatial pattern detected in previous projections (Meier et al., 2012b), with pronounced warming during the summers in the northern Baltic Sea (see supplementary material).

Due to the projected increased runoff, the volume averaged salinity decreases in all scenario simulations at the end of the century (Fig.5). However, the differences between GCMs are substantial. In the regionalization of Model C (HadGEM2-ES), the largest salinity decline, i.e., that of about -1.5 g kg-1, in future relative to the historical period is found. In contrast, Model B (EC-EARTH) causes a slight increase in salinity until approximately 2030, and its difference between RCP 4.5 and RCP 8.5 is smaller than the results given by other models. Thus, the range of salinity changes is large, and consequently, uncertainties in salinity projections are greater than those in the temperature projections (see Section 3.3). For Model C, the emission scenario RCP 8.5 was not simulated because a runoff projection from E-HYPE was not available. Hence, the two ensembles of salinity projections shown in Fig.5 should not be used to compare uncertainties in RCP 4.5 and RCP 8.5 projections.

Although the absolute values of the changes in temperature and salinity vary between the two greenhouse gas emission scenarios, the shape of the average vertical profile does not change significantly (supplementary data), and strong stratification will still be one of the main characteristics of the future Baltic Sea.



### 3.2 Biogeochemical variables

Both changing nutrient loads and changing physical conditions have impacts on biogeochemical processes and nutrient cycling in the water column and sediments. In case of the Reference scenario, the model projects that ensemble mean, annual nutrient concentrations averaged for the entire Baltic Sea will change, between 1976-2005 and 2069-2098 under the RCP 4.5 scenario,

by -62% for ammonium, +10% for nitrate and -24% for phosphate (Fig.6, middle panel). Decreased phosphate concentrations result in decreased primary production (-13%) and nitrogen fixation (-20%). As nitrate is not completely consumed during the spring bloom due to lacking phosphate, nitrate concentration increases (+10%). Average oxygen concentration is projected to slightly decrease by about -1%, probably as a consequence of the increasing water temperature. Following the decrease in phosphate and primary production, hypoxic area is 9% smaller than during the historical period.

In BSAP, the even larger reduction in nutrient loads results in a considerable reduction in primary production (-44%), nitrogen fixation (-96%) and hypoxic area (-32%) under the RCP 4.5 scenario (Fig.6, upper panel) whereas in the Worst Case primary production (+2%), nitrogen fixation (+22%) and hypoxic area (-3%) remain either unchanged or increase under the same greenhouse gas emission scenario (Fig.6, lower panel). Hence, changes in nutrient supply, in particular phosphorus, control the long-term response of eutrophication, biogeochemical fluxes and oxygen conditions in the deep water.

Under the warmer RCP 8.5 scenario, the response of the biogeochemical cycles to changes in nutrient loads (BSAP, Reference, Worst Case) is similar compared to RCP 4.5 (Fig.6). The projected changes in temperature, salinity and other variables result in larger eutrophication, productivity and oxygen depletion. However, the impact of changing climate is more pronounced in case of high nutrient loads like the Worst Case scenario than in case of low nutrient loads like the BSAP scenario. For BSAP the differences between RCP 4.5 and 8.5 scenario simulations are smaller than for Worst Case. Hence, the response of bio-

geochemical cycles to warming climate under various nutrient load scenarios is nonlinear. This result, found in our ensemble study, becomes particularly noticeable by analyzing summer bottom oxygen and hydrogen sulfide concentrations (Fig.7). For BSAP the differences in summer bottom oxygen concentrations between RCP 4.5 and 8.5 scenarios at the end of the century are small. Hydrogen sulfide does not occur even in the deepest parts of the Baltic Sea. However, in the Worst Case scenario large areas of the sea bottom below the halocline depth suffer from hydrogen sulfide with considerably larger concentrations

in RCP 8.5 compared to RCP 4.5. Under the BSAP and RCP 4.5 and 8.5, projected hypoxic area is about -32 and -37% of present day hypoxic area on average, respectively (Fig.6). Hypoxic area is successively larger with increasing nutrient loads and increasing warming. In the combination of the Worst Case and RCP 8.5 scenarios, about 80% of the Baltic proper will have on average anoxic bottom conditions during summer. However, even in the latter scenario simulation hypoxic area is still slightly smaller or about the same as under present conditions (Figs.6 and 8).

Independent of the climate scenario, RCP 4.5 or 8.5, primary production and nitrogen fixation increases in future climate under the Worst Case scenario and decreases under the BSAP (Fig.6). Again, whether the response of the biogeochemical fluxes will be affected by changing climate depends on the nutrient loads. For instance, under the Worst Case scenario nitrogen fixation will increase by 22 and 56% in RCP 4.5 and RCP 8.5, respectively, whereas under the BSAP nitrogen fixation will vanish in both cases approximately.




In Fig.8 the temporal evolutions of primary production, nitrogen fixation and hypoxic area are shown. The standard deviation among the four ensemble members is large. However, at the end of the century the different results for the Worst Case or Reference scenario and the BSAP are clearly distinguishable.

### 3.3 Impact of global mean sea level rise

In the following section, we compare the results of the scenario simulations with those of the two sensitivity experiments with a 1 m higher mean sea level. At the end of the century, the volume averaged salinity in the experiments with 1 m higher mean sea level driven by Model A under the RCP 8.5 scenario is approximately $1.5 g k g^{-1}$ higher than that in the corresponding scenario simulation without changing the mean sea level (Fig.9). For comparison, in our ensemble, the ranges of projected salinities at the end of the century under both RCP 4.5 and 8.5 scenarios amount to approximately $2$ $1.5 g k g^{-1}$ (Fig.9). As wind fields do not change significantly (not shown), the range is mainly explained by differences in the projected river runoff. According to Meier and Kauker (2003), 50% of the past variability of the salinity in the Baltic Sea is explained by runoff variations. The remainder variability is explained by wind variations on short and long time scales.

The differences in the projected hypoxic areas at the end of the century between simulations with 1 m higher mean sea level and without changing mean sea level are less than 10%, indicating a modest sensitivity to mean sea level change (Fig.10). Hence, the results suggest that the differing future nutrient loads will dominate the uncertainties in the hypoxic area projections if the range of nutrient loads is defined by the Worst Case and BSAP scenarios (cf. Fig.6). In addition, the uncertainty caused by the global models is considerable and significantly higher than the uncertainty due to greenhouse gas emission scenarios (either RCP 4.5 or RCP 8.5).

### 4 Discussion

In this study, an ensemble of 21 scenario simulations driven by four different GCMs and two sensitivity experiments on sea level rise were performed by combining different future climate scenarios and nutrient load projections for the 21st century. The sensitivity experiments are not transient scenario simulations following Stocker et al. (2013) because the 1 m higher mean sea level was applied as being constant in time during 2006-2098. The reason for this experimental setup is that the Baltic Sea model RCO has a linearized free sea surface following Killworth et al. (1991) that does not permit long-term changes in the mean sea surface height (Meier et al., 1999). Hence, our experiments overestimate the effect of the increasing global mean sea level and, thus, overestimate the increasing salinity in the Baltic Sea (Figs.9 and 10). A 1 m higher mean sea level was chosen in our sensitivity experiments because this value is close to the high-end scenario simulation results at the end of the 21st century (Stocker et al., 2013). As projections of global mean sea level rise are rather uncertain (Stocker et al., 2013), the aim of our sensitivity experiments is only to illustrate a possible maximum effect of increasing global mean sea level that has

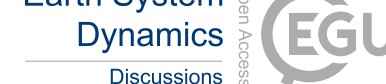



been neglected in all previous scenario simulations of the Baltic Sea (Meier et al., 2017).

The projected ensemble mean change in salinity amounts to -0.7 $1.5gkg^{-1}$ compared with that of the historical period under the RCP 4.5 scenario, with a considerable ensemble spread among the different GCMs. Under the RCP 8.5 scenario, the ensemble mean change of the three GCMs amounts to -0.6 $1.5gkg^{-1}$. However, since the ensemble excludes the model that shows the

greatest projected salinity change under the RCP 4.5 scenario (Model C), we assume that under RCP 8.5, the change of the ensemble mean will also be greater if Model C is included (Fig.9). Hence, our ensemble is too small and the uncertainties, inter alia, in the salinity projections, might be underestimated.

Substantial uncertainties in future projections for the Baltic Sea are caused by the driving GCMs, e.g., those for salinity due to uncertainties in projected river runoff and global mean sea level rise (Fig.9, c.f. Meier et al. (2017)). In RCP 4.5 and RCP

8.5, the projected runoff varies between 1 and 21 % and between 6 and 21 % among the models, respectively, explaining the considerable uncertainties in the projected salinity, which are much greater than the natural variability (Meier and Kauker (2003), their Fig.7). Uncertainties in salinity and stratification affect biogeochemical fluxes and hypoxic areas (Eilola et al., 2011). For instance, the vertical flux of oxygen between the well-oxygenated surface layer and the deep water is controlled by vertical stratification (Väli et al., 2013). Hence, deficiencies in GCMs have considerable impacts on the water balance of

the Baltic Sea that cannot be neglected in regional projections (Meier et al., 2011c). However, in our multi-model ensemble study, the resulting uncertainties in biogeochemical fluxes, primary production and nitrogen fixation, and hypoxic areas are still significantly smaller than the differences caused by the different nutrient loads scenarios, i.e., Worst Case and BSAP (2). In addition, the uncertainties caused by greenhouse gas emission scenarios (RCP 4.5 and 8.5) and global mean sea level rise are also smaller than the differences between nutrient load scenarios. Hence, we found an overwhelming impact of the

various nutrient load scenarios on the changing biogeochemical cycles in the Baltic Sea. For (1) primary production and (2) nitrogen fixation and hypoxic area, the second largest uncertainties are based on the choice of greenhouse gas emission scenario (RCP 4.5 or 8.5) and GCM deficiencies (calculated from four models), respectively (22). As one of the main uncertainties in the salinity projections is caused by the highly variable runoff projections between the GCMs (2, 9), we conclude that projections in nitrogen fixation and hypoxic area suffer from shortcomings in the simulated water cycles. For changes in primary production,

the magnitude of the temperature increase also plays an important role. In addition, the uncertainty in salinity changes due to global mean sea level rise has an important impact on nitrogen fixation and hypoxic area. The latter result is in agreement with Meier et al. (2017).

In this study, only results from greenhouse gas emission scenarios RCP 4.5 and 8.5 were analysed. RCP 2.6, at the lower end of the IPCC emission scenarios, corresponding to the goal of a global temperature rise limited to less than 2 °C, was not studied.

Hence, in our ensemble, the range of warming in the Baltic Sea region is smaller than that of the full range of global scenario simulations.

Further, we have not investigated the uncertainties caused by the shortcomings in the regional climate models of the Baltic Sea. Eilola et al. (2011) compared three different coupled physical-biogeochemical models for the Baltic Sea under the present climate conditions. This work concluded that the models reproduce much of the biogeochemical cycling in the Baltic proper

in hindcast simulations of 1970-2005. However, uncertainties caused by the assumptions about the bioavailable fractions of

nutrient loads from land and parameterizations of the key biogeochemical processes were considerable. The same models were also used in an ensemble of scenario simulations for 1960-2100 (Meier et al., 2011a, 2012a, c; Neumann et al., 2012). Within the latter studies, substantially differing nutrient loads scenarios and driving GCMs were applied, making a direct comparison with our results impossible. As only two driving GCMs were used and the impact of global mean sea level rise was neglected,

the previously published ensemble spread in salinity was smaller than that found in our study. Uncertainties in the projections of the hypoxic area were about as large as those presented in the ensemble of scenario simulations of this work. Hence, future projections of the Baltic Sea ecosystem require multi-model ensembles of regional and global climate models to be able to properly estimate uncertainties.

## 5   Conclusions

From the model results of this study we draw the following conclusions: (1) Implementation of the BSAP will lead to a significantly improved ecosystem state of the Baltic Sea irrespective of the driving GCM if changing climate does not counteract nutrient load reductions.

(2) The main driver of eutrophication is external nutrient loads. Climate change (mainly warming and global mean sea level rise) may amplify eutrophication. The response of biogeochemical fluxes, such as primary production and nitrogen fixation,

and deep water oxygen conditions to changing climate depend on the nutrient load scenario. In the case of high (low) nutrient loads, the impact of the changing climate would be considerable (negligible). However, the impacts of the changing climate within the range of the considered greenhouse gas emission scenarios (RCP 4.5 and 8.5) on biogeochemical cycles will be smaller than the impacts of the considered nutrient load changes (BSAP, Reference, Worst Case).

(3) Substantial uncertainties in the future projections for the Baltic Sea are caused by the driving GCMs, e.g., those for salinity

due to the uncertainties in the projected river runoff and global mean sea level rise. Hence, for future projections, an ensemble of various driving GCMs is necessary. This study also shows that dynamical downscaling is a useful tool because local drivers of marine biogeochemical cycling, such as nutrient load changes, are still more important than the estimated uncertainties caused by deficiencies of the global models. Despite the large uncertainties caused by global climate models, we were able to draw a conclusion concerning the impact of the BSAP in future climates (see conclusion no. 1).

*Code and data availability.*    Data sets and software code corresponding to the manuscript can be made available upon request.

## Appendix A:  Supplementary data

The next figures (Fig.A1,Fig.A2,Fig.A3) provide supplementary data to this manuscript.



*Competing interests.* no competing interests are present

*Acknowledgements.* The research presented in this study is part of the Baltic Earth program (Earth System Science for the Baltic Sea region, see http://www.baltic.earth) and was funded by the BONUS BalticAPP (Well-being from the Baltic Sea – applications combining natural science and economics) project which has received funding from BONUS, the joint Baltic Sea research and development programme (Art

5    185), funded jointly from the European Union´s Seventh Programme for research, technological development and demonstration and from Swedish Research Council for Environment, Agricultural Sciences and Spatial Planning (FORMAS, grant no. 942-2015-23). Additional support by FORMAS within the project "Cyanobacteria life cycles and nitrogen fixation in historical reconstructions and future climate scenarios (1850-2100) of the Baltic Sea" (grant no. 214-2013-1449) and by the CERES project, which has received funding from the European Union's Horizon 2020 research and innovation programme under grant agreement no. 678193, is acknowledged.



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



**Figure 1.** The Baltic Sea: bathymetry, river discharges, and sub-basins as defined in this study. Baltic proper comprises Arkona, Bornholm, East Gotland and Northwest Gotland basins. In addition, the location of the monitoring station at Gotland Deep (BY15) is shown (white circle).



**Figure 2.** Ensemble mean, monthly air temperature in 2m height (in °C) at Gotland Deep (BY15) calculated from four regional climate simulations: historical period (1976-2005) (solid black line) and future period (2069-2098) according to RCP 4.5 and RCP 8.5 scenarios (solid orange and red lines, respectively). The colored shaded areas denote the standard deviation among the ensemble members. In addition, observations and their standard deviations are shown (black symbols and vertical thin bars).



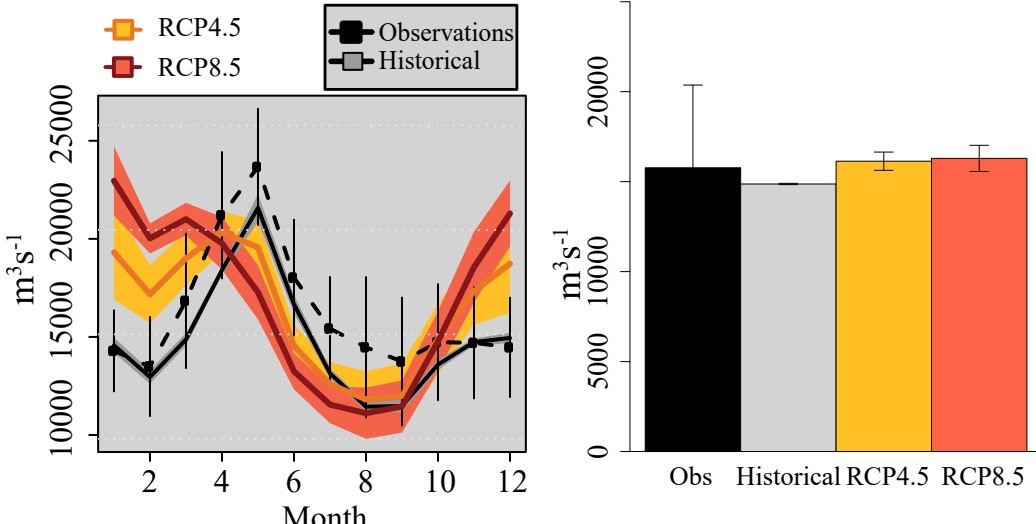

**Figure 3.** Ensemble mean monthly (left) and annual (right) runoff (in m3 s-1) for the Baltic Sea calculated from four hydrological model simulations: historical period (1976-2005) (solid black line in the left panel and grey bar in the right panel) and future period (2069-2098) according to RCP 4.5 (orange) and RCP 8.5 (red) scenarios, respectively. The colored shaded areas in the left panel denote the standard deviation among the ensemble members. In addition, observations and their standard deviations are shown (dotted black line and vertical thin bars in the left panel and black bar in the right panel).



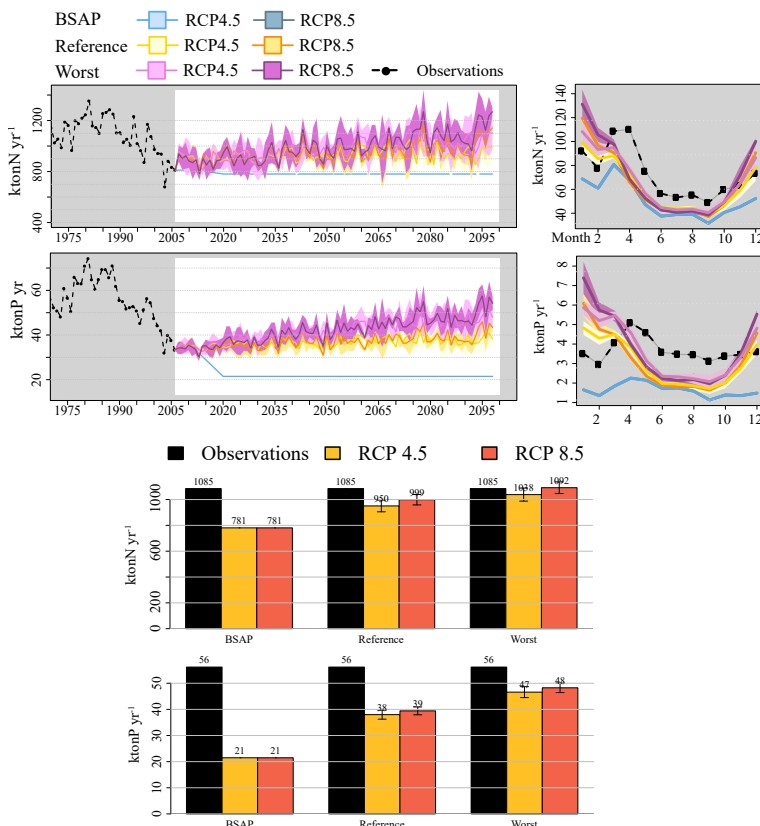

**Figure 4.** Observed and projected ensemble mean of the total bioavailable nutrient loads (nitrogen and phosphorus) to the Baltic Sea between 1970 and 2100 (upper, left panels), mean seasonal cycle (upper, right panels) and annual mean loads (lower). Shown is the sum of loads from rivers, point sources and atmosphere. Results were calculated from four hydrological model simulations during the historical (1976-2005) and future (2069-2098) periods according to the RCP 4.5 and RCP 8.5 scenarios combined with three nutrient loads scenarios (BSAP, Reference and Worst Case). The colored shaded areas denote the standard deviations among the ensemble members.



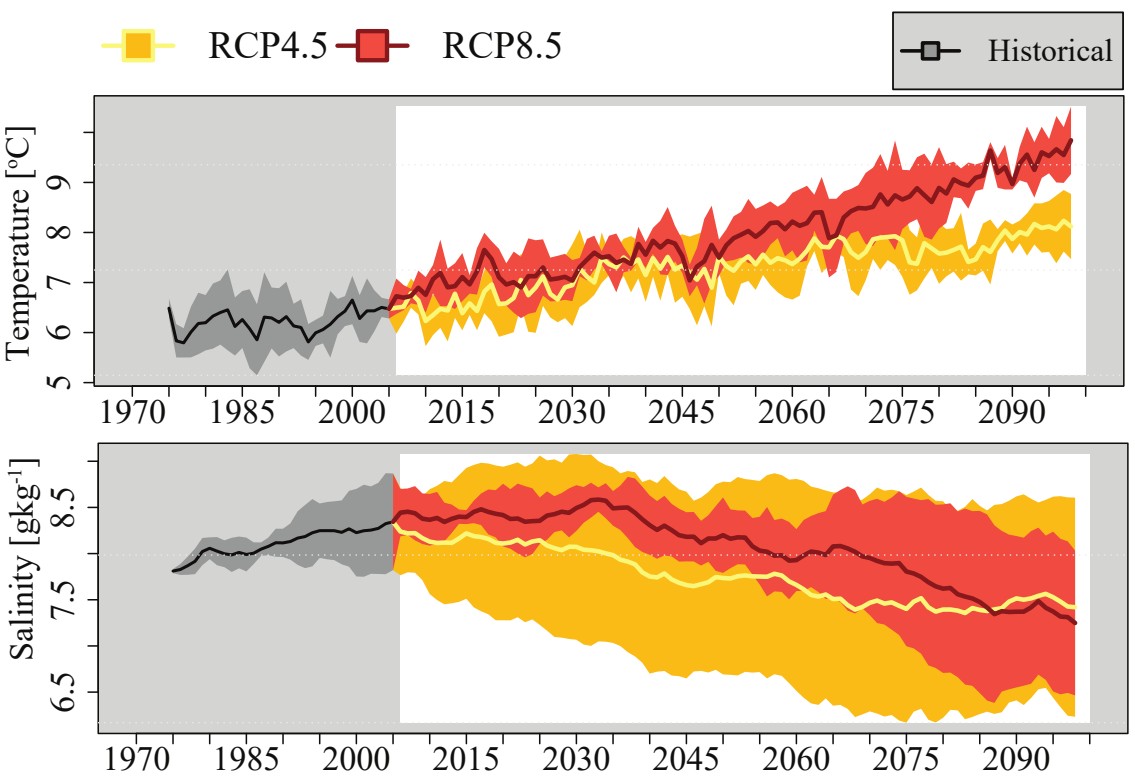

**Figure 5.** Ensemble mean volume averaged temperature (in °C) and salinity (in g kg-1) as a function of time for 1975-2098 in the two climate scenarios, RCP 4.5 (orange) and RCP 8.5 (red). The coloured shaded areas denote the standard deviations among the ensemble members.





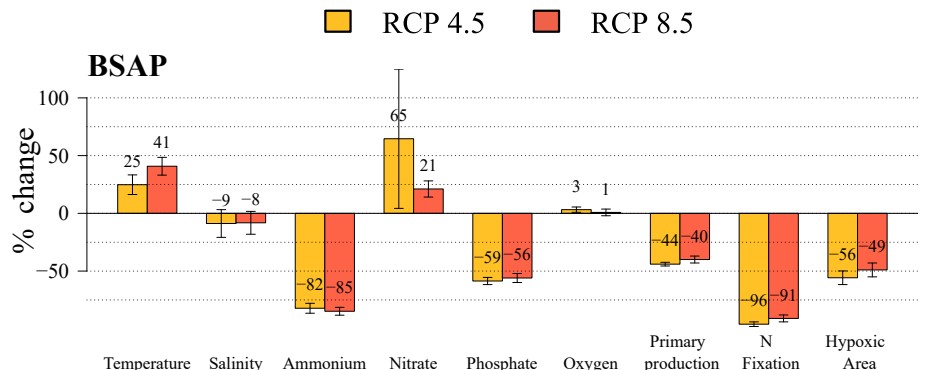

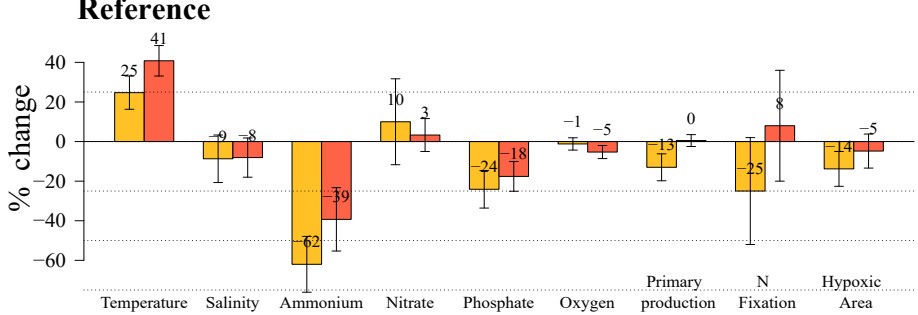

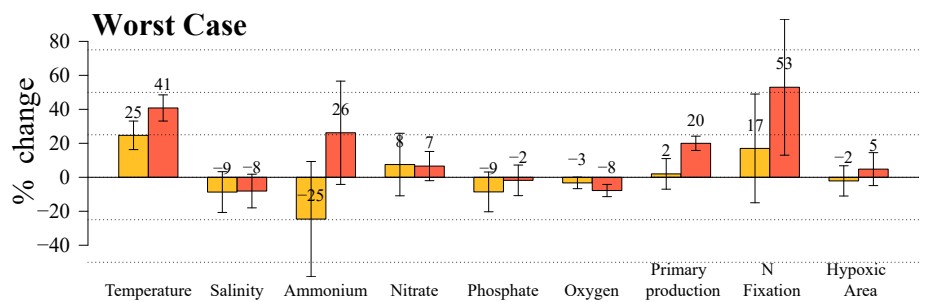

**Figure 6.** Relative ensemble mean volume averaged, 30-year mean changes between the future (2069-2098) and historical (1976-2005) periods in temperature, salinity, nutrient and oxygen concentrations, primary production, nitrogen fixation and hypoxic areas in the entire Baltic Sea under the different climate and nutrient load scenarios: BSAP (upper panel), Reference (middle panel) and Worst Case (bottom panel). The relative temperature changes are based on temperatures in °C. In addition, the standard deviation of changes among the ensemble members are shown.




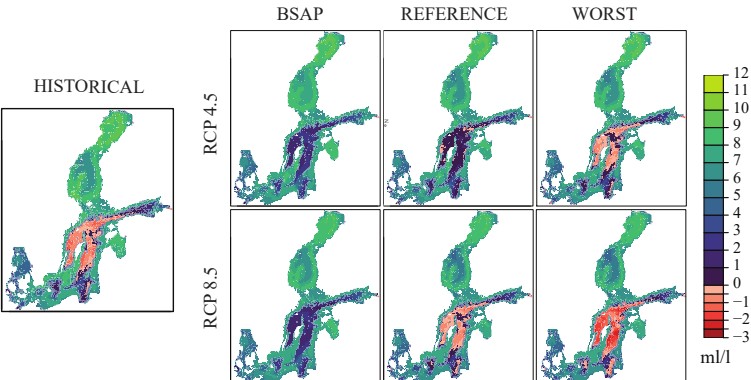

**Figure 7.** Historical (1976-2005) and projected future (2069-2098) ensemble mean summer bottom oxygen concentrations (in mL L-1) in three nutrient load (BSAP, Reference and Worst Case) and two greenhouse gas emission scenarios (RCP 4.5 and 8.5). Hydrogen sulfide concentrations are represented by negative oxygen concentrations (1 mL H2S L-1 = -2 mL O2 L-1).





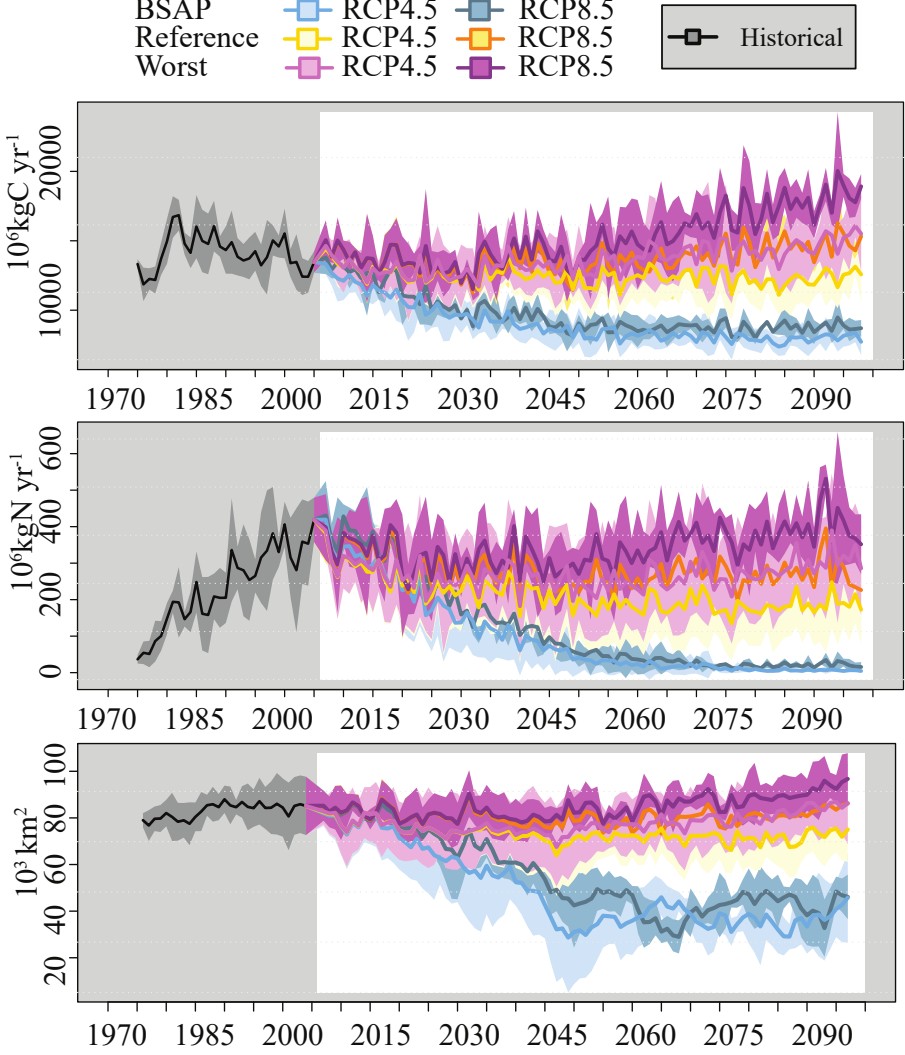

**Figure 8.** Temporal evolution of ensemble mean volume averaged primary production (in 106 kg C year-1, upper panel) and nitrogen fixation (in 106 kg N year-1, middle panel), and hypoxic area (in 103 km2, lower panel) in the entire Baltic Sea during 1975-2098 and their standard deviations (ensemble spread) among ensemble members. For all combinations of two greenhouse gas emission scenarios (RCP 4.5 and 8.5) and three nutrient load scenarios (BSAP, Reference and Worst Case) the ensemble mean and spread were calculated from four regionalized global climate simulations.





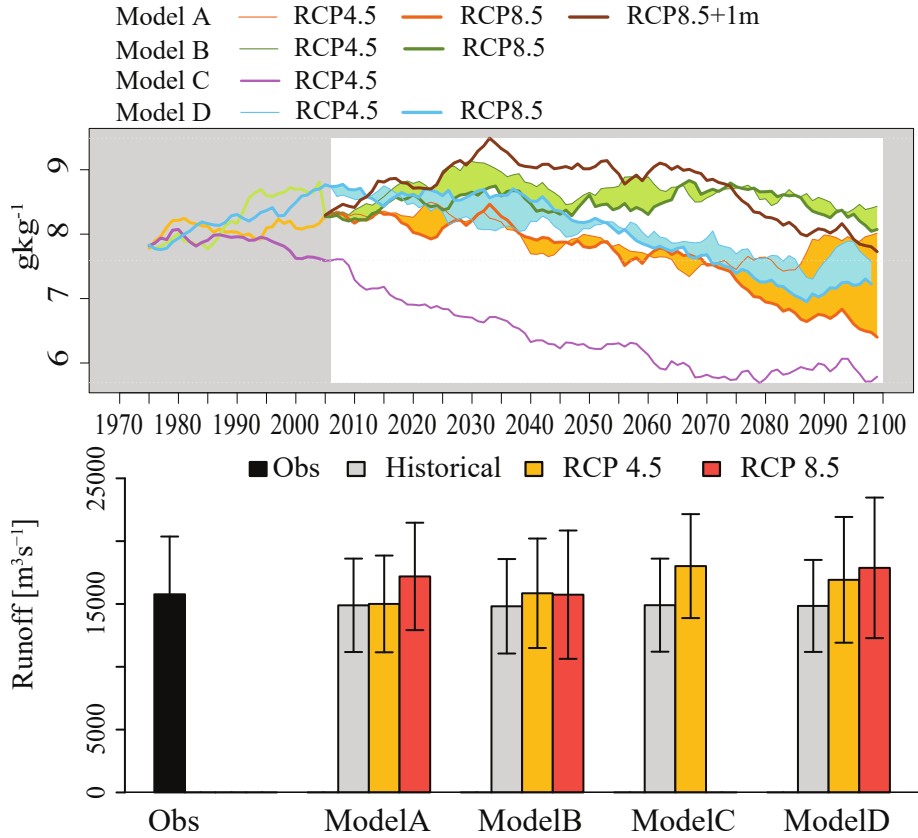

**Figure 9.** Upper panel: Temporal evolution of volume averaged salinity (in g kg-1) in the entire Baltic Sea during 1975-2098 in various climate scenarios (RCP 4.5 and 8.5) using four global climate models: MPI-ESM-LR (Model A); EC-EARTH (Model B); IPSL-CM5A-MR (Model C); HadGEM2-ES (Model D). Note, a scenario simulation driven by Model C and RCP 8.5 does not exist. A sensitivity experiment forced by Model A under the RCP 8.5 scenario assuming a 1 m higher mean sea level is also shown. Lower panel: Annual mean river runoff (in m3 s-1) in the scenario simulations and observations (1976-2005).





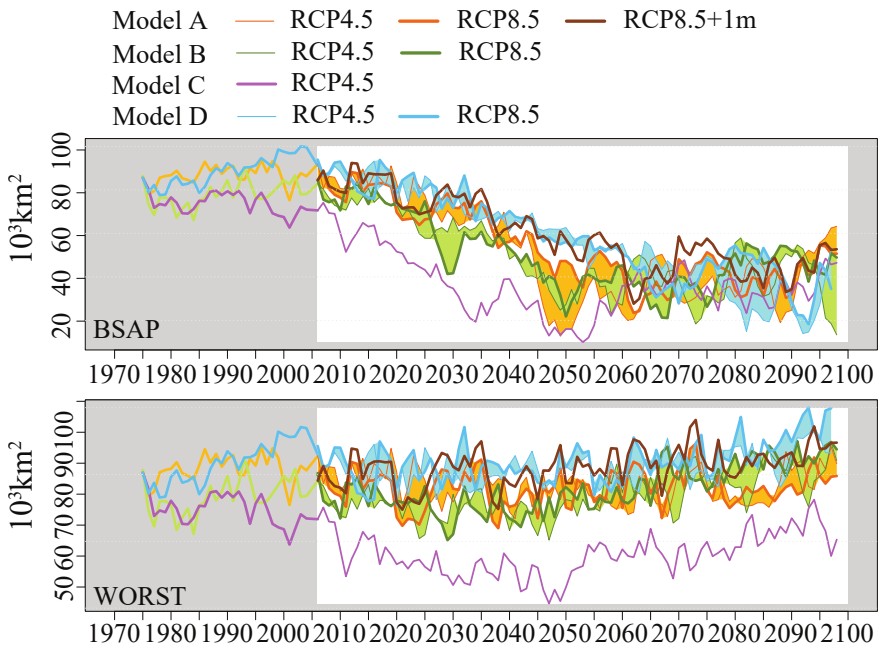

**Figure 10.** As the upper panel in Fig.9 but for hypoxic area under the BSAP and Worst Case scenarios.





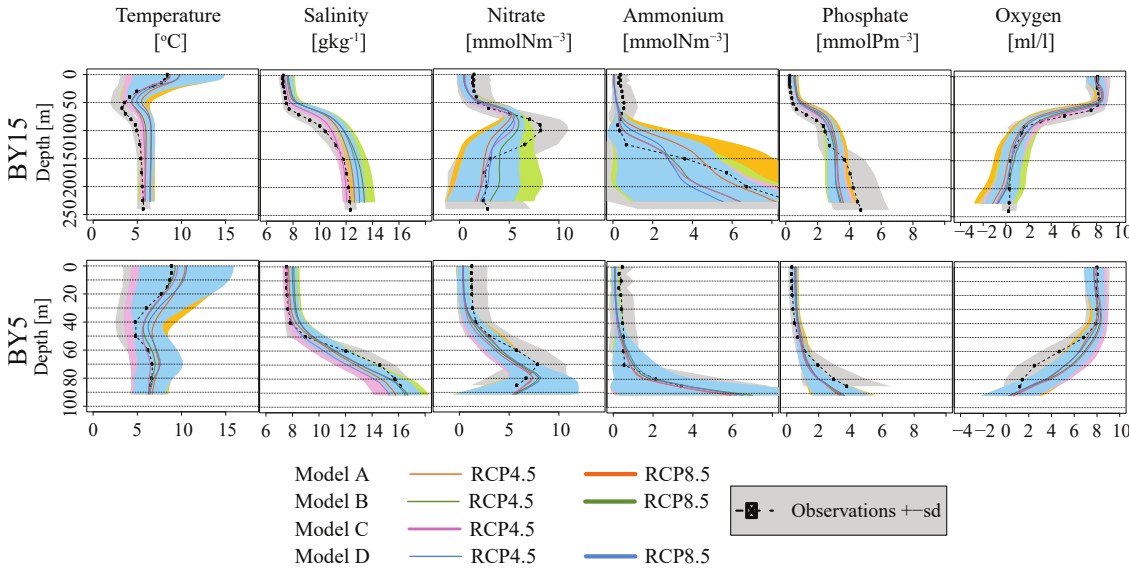

**Figure A1.** Simulated (solid colored lines) and observed (black dotted line) mean profiles of temperature, salinity, and nutrient and oxygen concentrations at the monitoring stations BY5 and BY15 for the historical (1976-2005) period. GCMs used are: MPI-ESM-LR (Model A); EC-EARTH (Model B); IPSL-CM5A-MR (Model C); HadGEM2-ES (Model D). Observations from BED are given at HELCOM standard depths and linearly interpolated between these depths. For observations (grey) and climate model results (color) ranges between minus and plus one standard deviation of the two-daily time series are shown as shaded areas.





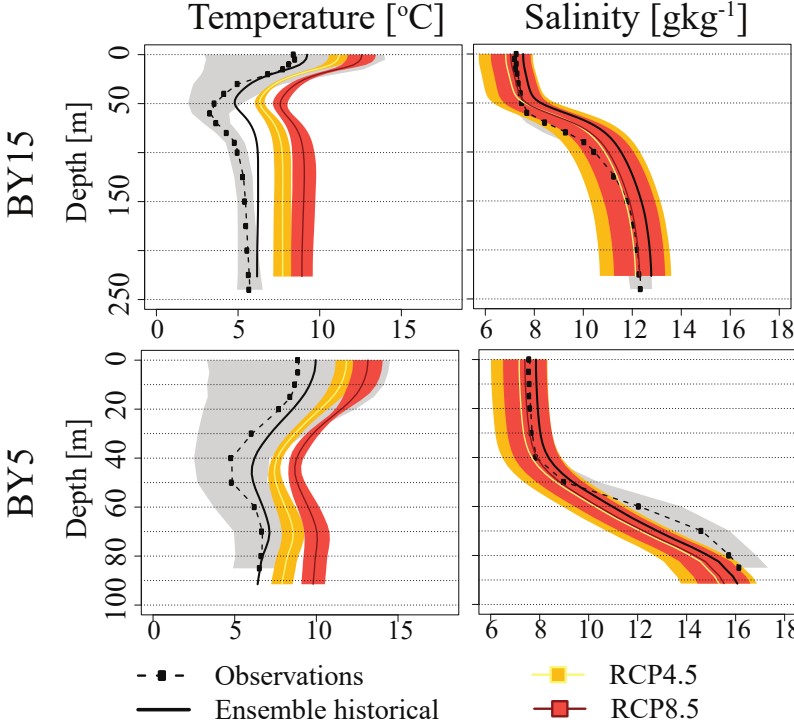

**Figure A2.** Projected (2069-2098) ensemble mean of temperature and salinity profiles assuming RCP 4.5 (light orange) and RCP 8.5 (dark orange), simulated ensemble mean of the historical period (1961-2005) (solid black line), and mean observations (dotted line). The colored shaded areas denote the standard deviation among the ensemble members.



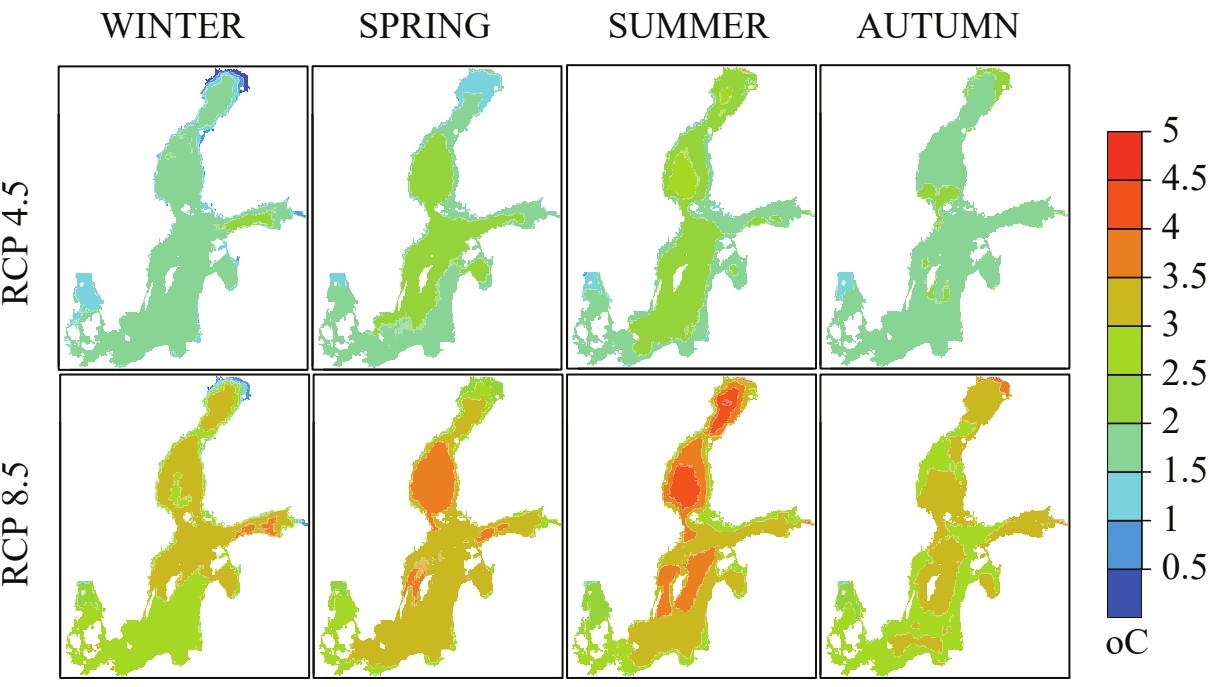

**Figure A3.** Projected changes in seasonal mean sea surface temperatures (in °C) in future climates according to RCP 4.5 (upper panels) and RCP 8.5 (lower panels).



**Table 1.** List of experiments (for details see text).

|  | BSAP | Reference | Worst Case |
|---|---|---|---|
| RCP 4.5 | Model A-D | Model A-D | Model A-D |
| RCP 8.5 | Model A, B, D | Model A, B, D | Model A, B, D |
| RCP 8.5 | Model A + 1 m |  | Model A + 1 m |

**Table 2.** Uncertainties expressed as standard deviations in temperature and salinity and variances of 30-year mean changes between the future (2069-2098) and historical (1976-2005) climates in primary production, nitrogen fixation and hypoxic area caused by GCMs, RCPs, nutrient loads, and global mean sea level rise (only model A, BSAP and Worst Case, see 1). Variances of changes in the primary production, nitrogen fixation and hypoxic area are normalized by the corresponding variances based on the changes in all 23 simulations.

| Parameter/Uncertainty | GCMs | RCPs | Nutrient loads | Sea level rise |
|---|---|---|---|---|
| Temperature (in $^{\circ}$C) | 0.5 | 0.8 | 0 | 0 |
| Salinity (in $gkg^{-1}$) | 0.9 | 0.4 | 0 | 1.1 |
| Primary production (in %) | 6 | 12 | 67 | 2 |
| Nitrogen fixation (in %) | 16 | 5 | 67 | 9 |
| Hypoxic area (in %) | 12 | 3 | 74 | 6 |