# Peer review of "Uncertainties in projections of the Baltic Sea ecosystem driven by an ensemble of global climate models"

_Earth System Dynamics, 2018_

## Referee Comment (RC1) · Anonymous Referee #1 · 29 Apr 2018

This article attempted to explore the uncertainties of biogeochemistry component, such as primary production, nitrogen fixation, and hypoxia area by using numerical experiments within Baltic Sea region. These uncertainties then discussed by various numerical scenarios among the change of nutrient loads, global model deficiencies and future greenhouse gas scenarios.

Although the context looks well fit the ESD journal, the writing skills of authors made a hard time to me to read through the entire paper. In such case, I would like to suggest giving the rejection or requesting the major revisions in the way of the writing first.

Here are some examples I would like to give.

[Figure]

• The abstract mentioned a jargon "transient simulations", which also exist within the introduction section; however, the authors didn't declare/define what it is.

• Line 22 on page 2 addressed "By approximately 18 years ago" to begin the sentence, but the sentence or the sentences nearby did not mentioned the reference year of "18 years ago". As a reader, it is hard to capture what the period of "18 years ago" is meaning? Is it the 18 year ago of 2018, 2017, 2006 or what?

• The authors keep using "GCM" or "GCMs", but didn't mentioned/defined what GCMs are. For example, line 25 on page 2, "The first scenario simulations were based on only one GCM and one greenhouse gas emission scenario, .... to future climates utilizing mini-ensembles consisting of two GCMs and two emission scenarios .... even used three GCMs and three emission scenarios."

This is hard to understand what "GCMs" and what "scenarios" the previous studies worked. It is also difficult to realize the thoughts of authors why they want to mention those of previous works, even though reader can "guess" the authors want to address the uncertainties issue from the usage of GCMs use.

• The name of section "Methods" should use "data and methodologies" instead, because you are only just saying what methods you will use but also the data you apply.

• Again, it is confused by the abbreviations authors used. For example, line 17 on page 4 saying "The Baltic Sea model was forced by (1) atmospheric surface fields from a regional coupled atmosphere-ocean climate model (RCM) driven by lateral boundary conditions from GCMs and by (2) runoff and (3) nutrient loads from a regional hydrological model also forced by regionalized atmospheric data from the same GCMs. The RCM is RCA4-NEMO applied to the EURO-CORDEX domain with an interactively coupled Baltic Sea and North Sea."

Is it necessary to give the abbreviation "RCM" to use in the sentence? Is it possible

to just use RCA4-NEMO to instead? What is RAC4-NEMA? What is EURO-CORDEX domain? If RCA4-NEMO is coupled model, why you only briefly explain the RCA4 but no description for the ocean component? Also, the description is too shallow to understand what it is.

 c There is no connection between sentences and sentences. For example, in the line 21 on page 4, "RAC4 has a 0.22-degree spherical rotated latitude/longitude grid with 40 vertical levels. The hydrological model is E-HYPE, a process-based multi model applied for Europe. The runoff from each river was corrected by a factor . . . . . . "

From these few sentences, it is understood that there is an atmospheric component called RCA4, but there is no clue saying that the E-HYPE is the ocean component of the coupled model.

 c Line 24 on page 4. "The runoff from each river was corrected by a factor that corresponds to the ratio . . . This approach has been previously applied for regional climate simulations of the Baltic Sea."

Authors did not explain it clearly why they want to use this ratio to correct the river runoff. In addition, the sentence I addressed above didn't have the connection with the sentence below "here, improved versions of the regional and global climate models and the results of scenario simulations from the latest IPCC assessment report were used." It is hard to understand the authors want to speak out again.

 c Line 30 on page 4. "RCP 4.5 and 8.5 are medium and high-end scenarios, respectively." As a reader, it is no sense to know what are RCP4.5 and RCP8.5, what those scenarios look like. This is the duties of authors to declare their words clearly rather than the readers guess the authors' thoughts.

 c Line 1 on page 5. "The selection of these GCMs follows the approach presented by Wilcke and Barring (2016)". Again, what is the "approach" you are saying it. You need to explain it.

[Figure]

• Line 4 on page 5. "The four driving GCMs of this study were selected from the distinct clusters identified y Wilcke and Barring (2016)." Is this redundant sentence comparing with line 1 on page 5?

• Line 5 on page 5. What is "necessary/unnecessary lateral boundary data"? Meanwhile it is little bit confused about the selected GCMs came from. It is "Wilcke and Barring" or "SMHI".

• Line 7 to Line 11 on page 5. 1. High quality reanalysis dataset EUR04M. What is the definition of "high quality"? Is ERA40 low quality reanalysis? Than why do not compare with ERA-Interim or other ECMWF products? Or what do not you to compare with satellite observed winds? 2. Where is the correct factor 1.6 coming from? Why is 1.6? Why is not 2.0, 3.0, or 4.0? You didn't mention it.

• Line 12 to Line 18 on page 5. 1. You were mentioned the wind correction of RCA4-NEMO output, and then jump to an example. Normally, this example should be the example of your wind correction; however, you are addressing the "air temperature" and "total runoff" as examples. It is hard to see the connection between the sentences. 2. Meanwhile, what air temperature you are using? On the surface layer? On the 850mb, or on 200mb? 3. Since the previous description did not clearly describe the RCP 4.5 and RCP 8.5, it is hard for readers to understand what "present" and "future" climates in your figures. 4. When you use legend for your figures, you want to keep consistency between your figures and legends. You are using line with square in the legend, then should use line with square in the plots. 5. You are addressing the annual men for both air temperature and runoff. But you can only see the runoff annual mean plots existed, but no for air temperature. Where is it? Why do not you to show it?

• Subsection of nutrient load scenarios. 1. It is understood that there is three nutrient loads scenarios are applied in your study, but it is hard to understand why you want to pick up this three scenarios. It is better to describe the purposes and differences between each nutrient scenario clearly. 2. Line 34 on page 5 and Line 1 on page 6.

"are multiplied by a factor that summarizes the impact of a worst case socio-economic development on current nutrient loads. The main assumptions and description of this impact factor can be found in (Zandersen et al., in prep)."

Here is the thing. You didn't explain why you want to multiple a factor. You didn't explain how did you obtain this factor. You didn't address the assumptions and description of this impact factor but addressing a non-published article. How does the reader know what/why/how you are doing this? 3. Line 3 on page 6. "as well. In all three scenarios, nutrient loads . . .." The sentence and the previous sentence are not talking about the exactly the same thing. It is necessary to give a line break here.

• After read through the entire method section, It is difficult to understand that the authors are running regional coupled model – RCA4-NEMO or running the Baltic Sea model or running the four GCMs (MPI-ESM-LR, EC-EARTH, IPSL-CM5a-MR, and HadGEM2-ES) to conduct their research.

Please revise your writing for the reviewer. Good luck.

---

## Author Comment (AC1) · 4 Jun 2018

[12pt]article graphics

**Answers to the comments submitted by anonymous Referee #1 concerning the manuscript entitled "Uncertainties in projections of the Baltic Sea ecosystem driven by an ensemble of global climate models" by Sofia Saraiva et al.**

We acknowledge the comments of the reviewer who requested additional explanations

of specific terms used in climate modelling rather than language issues. We rephrased large parts of the text and added a table that explains acronyms (Tab. 1). We hope that the manuscript is now better readable for the reviewer and the readership of ESD. The answer to specific comments are addressed bellow and a new version of the manuscript can be found as a supplement to this comment.

*This article attempted to explore the uncertainties of biogeochemistry component, such as primary production, nitrogen fixation, and hypoxia area by using numerical experiments within Baltic Sea region. These uncertainties then discussed by various numerical scenarios among the change of nutrient loads, global model deficiencies and future greenhouse gas scenarios. Although the context looks well fit the ESD journal, the writing skills of authors made a hard time to me to read through the entire paper. In such case, I would like to suggest giving the rejection or requesting the major revisions in the way of the writing first. Here are some examples I would like to give.*

1. *The abstract mentioned a jargon "transient simulations", which also exist within the introduction section; however, the authors didn't declare/define what it is.*
   **Answer:** The concept of transient simulations is common within the climate modelling community in contrast to the time slice approach. Some time ago, for several reasons, models were not able to perform continuous simulations of 100 years. Thus, the modelling approaches were based on time slices: the model would run for about 30 years in the present climate and for another 30 years in the future. That approach was the possible one, but it had several faults (that we will not explore now but for example the initialization of the future period). With the increase in computational resources this limitation does not exist anymore, and we were able today to perform continuous, transient simulation. To avoid confusion, we omitted 'transient'.

2. *Line 22 on page 2 addressed 'approximately 18 years ago' to begin the sentence, but the sentence or the sentences nearby did not mentioned the reference year of '18 years ago'. As a reader, it is hard to capture what the period of '18 years ago' is meaning? Is it the 18 year ago of 2018, 2017, 2006 or what?*
**Answer:** We agree and changed the text accordingly.

3. *The authors keep using "GCM" or "GCMs", but didn't mentioned/defined what GCMs are. For example, line 25 on page 2, "The first scenario simulations were based on only one GCM and one greenhouse gas emission scenario, .... to future climates utilizing mini-ensembles consisting of two GCMs and two emission scenarios .... even used three GCMs and three emission scenarios." This is hard to understand what "GCMs" and what "scenarios" the previous studies worked. It is also difficult to realize the thoughts of authors why they want to mention those of previous works, even though reader can "guess" the authors want to address the uncertainties issue from the usage of GCMs use.*
**Answer:** GCM stands for General Circulation model and was defined in the abstract and in the introduction when GCM is mentioned for the first time.

4. *The name of section "Methods" should use "data and methodologies" instead, because you are only just saying what methods you will use but also the data you apply.*
**Answer:** We changed the title of the sub-section.

5. *Again, it is confused by the abbreviations authors used. For example, line 17 on page 4 saying "The Baltic Sea model was forced by (1) atmospheric surface fields from a regional coupled atmosphere-ocean climate model (RCM) driven by lateral boundary conditions from GCMs and by (2) runoff and (3) nutrient loads*

*from a regional hydrological model also forced by regionalized atmospheric data from the same GCMs. The RCM is RCA4-NEMO applied to the EURO-CORDEX domain with an interactively coupled Baltic Sea and North Sea." Is it necessary to give the abbreviation "RCM" to use in the sentence? Is it possible to just use RCA4-NEMO to instead? What is RAC4-NEMA? What is EURO-CORDEX domain? If RCA4-NEMO is coupled model, why you only briefly explain the RCA4 but no description for the ocean component? Also, the description is too shallow to understand what it is.*

**Answer:** We believe that it is important to specify that our methodology uses results from a coupled atmosphere-ocean RCM model (regional climate model) because that is in fact a distinction from other studies. We realize that the high number of abbreviations can be difficult to read but most of them (e.g. EURO-CORDEX or NEMO model) are very common within the climate science community. Also, considering that the manuscript is already long, we made the option to describe briefly the used models, though providing references for a more detailed description. Nevertheless, the subsection on 'Regional climate data sets' of the manuscript was reformulated and expanded to clarify the description of the different models.

6. *There is no connection between sentences and sentences. For example, in the line 21 on page 4, "RAC4 has a 0.22-degree spherical rotated latitude/longitude grid with 40 vertical levels. The hydrological model is E-HYPE, a process-based multi model applied for Europe. The runoff from each river was corrected by a factor : : :: : :" From these few sentences, it is understood that there is an atmospheric component called RCA4, but there is no clue saying that the E-HYPE is the ocean component of the coupled model.*

**Answer:** We hope that the rephrased sentences are now clearer in the manuscript.
7. *Line 24 on page 4. "The runoff from each river was corrected by a factor that corresponds to the ratio : : : This approach has been previously applied for regional climate simulations of the Baltic Sea." Authors did not explain it clearly why they want to use this ratio to correct the river runoff. In addition, the sentence I addressed above didn't have the connection with the sentence below "here, improved versions of the regional and global climate models and the results of scenario simulations from the latest IPCC assessment report were used." It is hard to understand the authors want to speak out again.*
**Answer:** Following the previous remark, the sentences are rephrased. The runoff from the E-HYPE model was corrected so that the model results match the observations in the historical period. This decision was made after realizing that the bias in the E-HYPE model could affect significantly the hydrodynamic results.

8. *Line 30 on page 4. "RCP 4.5 and 8.5 are medium and high-end scenarios, respectively." As a reader, it is no sense to know what are RCP4.5 and RCP8.5, what those scenarios look like. This is the duties of authors to declare their words clearly rather than the readers guess the authors' thoughts.*
**Answer:** RCP 4.5 and RCP 8.5 are common abbreviations for the two most relevant Representative Concentration Pathways adopted by the IPCC and its detail description would be out of scope of the manuscript. For that reason, we invite the reader to follow the references provided in the text. Nevertheless, the text was expanded.

9. *Line 1 on page 5. "The selection of these GCMs follows the approach presented by Wilcke and Barring (2016)". Again, what is the "approach" you are saying it.*

*You need to explain it.*
**Answer:** For clarification we rephrased the text. The approach by Wilcke and Bärring (2016) is now briefly described: 'The selected models agree with the results obtained by Wilke and Bärring (2016) for the regional climate systems of the North Sea and Baltic Sea, which used hierarchical clustering methods to select an optimum subset of models, from the entire ensemble, to estimate uncertainties inherent in an ensemble with a minimum number of simulations.'

10. Line 4 on page 5. "The four driving GCMs of this study were selected from the distinct clusters identified by Wilcke and Bärring (2016)." Is this redundant sentence comparing with line 1 on page 5?
    **Answer:** Yes, the reviewer's comment is correct. The sentence was redundant and it was changed.

11. *Line 5 on page 5. What is "necessary/unnecessary lateral boundary data"? Meanwhile it is little bit confused about the selected GCMs came from. It is "Wilcke and Barring" or "SMHI".*
    **Answer:** The sentence was rephrased.

12. *Line 7 to Line 11 on page 5. 1. High quality reanalysis dataset EUR04M. What is the definition of "high quality"? Is ERA40 low quality reanalysis? Than why do not compare with ERA-Interim or other ECMWF products? Or what do not you to compare with satellite observed winds? 2. Where is the correct factor 1.6 coming from? Why is 1.6? Why is not 2.0, 3.0, or 4.0? You didn't mention it.*
    **Answer:** The reviewer's comment is correct. 'high quality' term was removed from the text. The factor of 1.6 comes from comparing the winds at selected positions over sea in respective dataset. When the wind speeds are plotted

against each other (sorted individually) they form a rather straight line with a knee at 10 m/s and a slope above the knee that indicates a factor of 1.6 wrong. The factor 1.6 comes from manual fitting of a line. We can say that is not 2.0 but we can not say if it should be 1.61 instead. We have extended the text to explain this but without going into any details.

13. *Line 12 to Line 18 on page 5. 1. You were mentioned the wind correction of RCA4-NEMO output, and then jump to an example. Normally, this example should be the example of your wind correction; however, you are addressing the "air temperature" and "total runoff" as examples. It is hard to see the connection between the sentences.*
    **Answer:** The reviewer is correct. The link between the sentences was perhaps not the best and the wording has been improved in the revised version.

14. *Meanwhile, what air temperature you are using? On the surface layer? On the 850mb, or on 200mb?*
    **Answer:** We analysed the 2 m air temperature which is a diagnostic output variable of atmospheric models. This is added to the text.

15. *Since the previous description did not clearly describe the RCP 4.5 and RCP 8.5, it is hard for readers to understand what "present" and "future" climates in your figures.*
    **Answer:** RCP 4.5 and RCP 8.5 are now clearly described in the manuscript as the Representative Concentration Pathways adopted by the IPCC in the latest assessment report.

16. *When you use legend for your figures, you want to keep consistency between*

*your figures and legends. You are using line with square in the legend, then should use line with square in the plots.*

**Answer:** We changed the text accordingly.

17. *You are addressing the annual men for both air temperature and runoff. But you can only see the runoff annual mean plots existed, but no for air temperature. Where is it? Why do not you to show it?*

    **Answer:** We followed the suggestion by the reviewer and added the information about spatially averaged air temperature changes in RCP 4.5 and RCP 8.5 to the text. As we do not have observations we have not displayed the two numbers as bar plot.

18. *Subsection of nutrient load scenarios. It is understood that there is three nutrient loads scenarios are applied in your study, but it is hard to understand why you want to pick up this three scenarios. It is better to describe the purposes and differences between each nutrient scenario clearly.*

    **Answer:** Changes were made in the description of the three scenarios that span a range of plausible future socio-economic conditions from the most optimistic to the worst scenario. The definition of the scenarios it's in fact quite strait forward: (1) one scenario that reflects the most optimistic possibility: the application of the BSAP; (2) the reference, meaning a scenario where nothing from the socio-economic point of view is changed (cities location, land use, industry) and the changes are only driven by climate changes; and the worst scenario were we assume not only climate change but also the worst socio-economical development, meaning the one with absolutely no care about the ecological impacts of socio-economic growth. We agree with the reviewer, that the definition of these scenarios becomes clearer by showing the differences between them. This is the reason why Fig. 4 is presented.

19. Line 34 on page 5 and Line 1 on page 6. "are multiplied by a factor that summarizes the impact of a worst case socio-economic development on current nutrient loads. The main assumptions and description of this impact factor can be found in (Zandersen et al., in prep)." Here is the thing. You didn't explain why you want to multiple a factor. You didn't explain how did you obtain this factor. You didn't address the assumptions and description of this impact factor but addressing a non-published article. How does the reader know what/why/how you are doing this?

**Answer:** Changes were made in the manuscript to clarify this topic. However, the methodology to estimate the socio-economic impact factor is not detailed described in this manuscript, as it was not performed by the authors and its complexity deserves the reading of the referenced paper. Nevertheless, we have changed the manuscript to include some more details on what the impact factor is, referring that it takes into account the changes induced in the nutrient loads and atmosphere deposition by the adoption of each SSP (shared socio-economic pathways) in the Baltic Sea region.

20. Line 3 on page 6. "as well. In all three scenarios, nutrient loads : : :." The sentence and the previous sentence are not talking about the exactly the same thing. It is necessary to give a line break here.

**Answer:** A line break was added.

21. *After read through the entire method section, It is difficult to understand that the authors are running regional coupled model – RCA4-NEMO or running the Baltic Sea model or running the four GCMs (MPI-ESM-LR, EC-EARTH,*

*IPSL-CM5a-MR, and HadGEM2-ES) to conduct their research.*

**Answer:** We have used results from the regional climate model RCA4-NEMO and results from the land surface model E-HYPE that both were run using four GCMs and two greenhouse emission scenarios to force a Baltic Sea model. We have revised the text to explain the methodology better.

**Supplement:**

**Uncertainties in projections of the Baltic Sea ecosystem driven by an ensemble of global climate models**

Sofia Saraiva1,3, H.E. Markus Meier2,3, Helén Andersson3, Anders Höglund3, Christian Dieterich3, Robinson Hordoir3, and Kari Eilola3

[revised manuscript text omitted]
., 2010). RCPs are greenhouse gas concentration trajectories adopted by the Intergovernmental Panel of Climate Change (IPCC) for its fifth Assessment Report (AR5) in 2013 (Stocker et al., 2013). The RCPs are named after the radiative forcing values in the year 2100 relative to pre-industrial values, i.e., +4.5 and +8.5 W m-2, respectively).
  - 6. improved versions of the coupled physical-biogeochemical model of the Baltic Sea with the aim of reducing model shortcomings;
  - 7. improved versions of the global models from the Coupled Model Intercomparison Project 5 (CMIP5) of the IPCC (Stocker et al., 2013).
- 25 The paper is organized as follows. In Section 2, the regional climate ocean model, the regional coupled atmosphere-ocean model, driving GCMs, greenhouse gas concentration and nutrient load scenarios and the experimental setup are introduced. In Section 3, the results of future projections for temperature, salinity, selected biogeochemical fluxes (primary production and nitrogen fixation) and hypoxic areas are presented. In Section 4, the suspected shortcomings of the study are discussed. In Section 5, some conclusions of the study are drawn.
- 30

20

**2 Data and Methodologies**

A series of scenario simulations with a coupled physical-biogeochemical Baltic Sea model (Section 2.1) was performed. The Baltic Sea model was driven by regionalized GCM data using the dynamical downscaling approach (Section 2.2). In this approach, the atmospheric forcing data were calculated using a RCM with GCM data at the lateral boundaries. The resulting

- 5 atmospheric surface fields were then applied to force the Baltic Sea model mentioned above and a hydrological/land surface model for the Baltic Sea catchment area. The output variables of the latter model are river runoff and nutrient loads to the Baltic Sea model. As the nutrient loads depend not only on precipitation and air temperature at the land surface, but also on land use, agricultural practices and sewage water treatment, all scenario simulations were performed under different socioeconomic scenarios covering a plausible range between low and high loads (Section 2.3). As the global mean sea level rise is
- 10 not considered in all scenario simulations, two additional sensitivity experiments were performed to estimate the impact of a higher sea level at the lateral boundary of the Baltic Sea model (Section 2.4).

**2.1 Baltic Sea Model**

In this study, a three-dimensional ocean circulation model is used in climate simulations for the period 1975-2098. RCO-SCOBI consists of the physical Rossby Center Ocean (RCO) (Meier et al., 2003; Meier, 2007) and the Swedish Coastal and Ocean

- 15 Biogeochemical (SCOBI) models (Eilola et al., 2009; Almroth-Rosell et al., 2011). The model domain covers the Baltic Sea area with an open boundary in the northern Kattegat (Fig.1). The horizontal and vertical resolutions are 3.7 km and 3 m (corresponding to 83 depth levels), respectively. In the water column, the biogeochemical model SCOBI describes the dynamics of nitrate, ammonium, phosphate, three phytoplankton groups (diatoms, flagellates and others, and cyanobacteria), zooplankton, detritus, oxygen and hydrogen sulfide as negative oxygen equivalents  $(1mLH_2SL^{-1} = -2mLO_2L^{-1})$ . In the present version,
- 20 the nitrogen and phosphorus detritus were separated according to (Savchuk, 2002). The sediment contains nutrients in the form of benthic nitrogen and benthic phosphorus. With the help of a simplified wave model, the resuspension of organic matter is calculated (Almroth-Rosell et al., 2011). RCO-SCOBI has previously been evaluated and applied in numerous long-term climate studies, e.g., Meier et al. (2003); Meier (2007); Meier et al. (2011a, 2012b); Eilola et al. (2009); Eilola et al. (2011); Almroth-Rosell et al. (2011); Schimanke and Meier (2016).
- 25

**2.2 Regional climate data sets**

The Baltic Sea model was forced by atmospheric surface fields from the coupled Rossby Center Atmosphere Version 4 and Nucleus for European Modelling of the Ocean models (RCA4-NEMO) applied to the EURO-CORDEX domain (Coordinated Downscaling Experiment for Europe, http://www.euro-cordex.net/) (Jacob et al., 2014) driven by lateral boundary con-

30 ditions from four GCMs. RCA4-NEMO is a regional coupled atmosphere-ocean climate model with an interactively coupled Baltic Sea and North Sea (Dieterich et al., 2006; Wang et al., 2015; Gröger et al., 2015), which allows a more realistic climate representation (Dieterich et al., submitted manuscript). The set of GCMs used in this study includes: MPI-ESM-LR (https://www.mpimet.mpg.de), EC-EARTH (https://www.knmi.nl), IPSL-CM5A-MR (http://icmc.ipsl.fr/) and HadGEM2-ES (http://www.metoffice.gov.uk), called Models A, B, C and D, respectively. This set is in agreement with the results obtained by Wilcke and Bärring (2016) for the climate systems of the North Sea and Baltic Sea regions, which used hierarchical clustering methods to select an optimum subset of models, from the entire ensemble, to estimate uncertainties inherent in an ensemble

- 5 with a minimum number of simulations. The lateral boundary data for RCA4-NEMO from each of these GCMs were provided
  - by the Rossby Center of the Swedish Meteorological and Hydrological Institute (SMHI).

Strong winds in the regionalized atmospheric forcing are underestimated compared to the reanalysis dataset EURO4M as well as earlier used regionalized forcing of ERA40 (Meier et al., 2011b). The latter uses a correction of strong winds based on gustiness, but this parameter was not available for the current regionalized forcing. The statistical distribution of the winds

10 over sea in the two reanalysis datasets (with the correction in the latter case) agrees well with each other. The winds in the current regionalized forcing agrees well with these two reanalysis data sets up to  $10 ms^{-1}$  but underestimates the winds above  $10 ms^{-1}$  by a factor of 1.6. Therefore, a correction was made by multiplying the portion of the wind exceeding  $10 ms^{-1}$  by 1.6 without altering the direction.

The river runoff and nutrient loads are results from the hydrological model E-HYPE (Hydrological Predictions for the En-

- 15 vironment, http://hypeweb.smhi.se), which is a process-based multi-basin model applied for Europe (Hundecha et al., 2016; Donnelly et al., 2013, 2017). To minimize uncertainties caused by the hydrological model bias (results not shown), the runoff from each river was corrected for the historical and future periods so that the total annual flow to the Baltic Sea estimated by the model matches the observations during the historical period (1971-2005). In this study, only results from greenhouse gas concentration scenarios RCP 4.5 and 8.5 were analyzed. RCP 2.6, at the lower end of the IPCC concentration scenarios,
- 20 corresponding to the goal of a global temperature rise limited to less than 2 °C, was not studied. Hence, in our ensemble, the range of warming in the Baltic Sea region is smaller than that of the full range of global scenario simulations. Figures 2 and 3 show the results of the seasonal cycles of regionalized 2 m air temperature over the central Baltic and the total runoff in present and future climates. In the future climate (2069-2098), air temperatures over the central Baltic will increase more in winter than in summer, and runoff from the entire catchment area will increase during winter but decrease during summer. In terms of
- 25 the annual mean averaged over the Baltic Sea, both temperature and runoff will increase in the future compared to those figures of the historical climate. During the historical period, annual and monthly biases of both variables were within the range of variability of the observations, i.e., within the range of plus or minus one standard deviation from the monthly mean.

**2.3 Nutrient load scenarios**

Climate projections for the Baltic Sea are carried out under the three nutrient load scenarios described below, spanning a range of plausible future socio-economic conditions from the most optimistic to the worst scenario. During the historical period (1976-2005), the observed nutrient loads from the Baltic Environmental Database (BED) are used (http://nest.su.se/bed/).

1. Baltic Sea Action Plan (BSAP) scenario (HELCOM, 2013). In this scenario, nutrient loads from rivers and atmospheric deposition in the different sub-basins will linearly decrease after 2012 from the current values (average 2010-2012)

as estimated by Svendsen et al. (2015) to the maximum allowable input defined by the BSAP until 2020. After 2020, nutrient loads will remain constant until 2098.

- 2. Reference scenario. In this scenario, E-HYPE projections for future nutrient loads (2006-2098) under the two different greenhouse gas concentration scenarios (RCP 4.5 and RCP 8.5) are used, assuming no socio-economic changes compared to the historical period (1976-2005). Hence, e.g., land and fertilizer usage, soil properties and sewage water treatment in each sub-basin do not change over time. Only the impacts of the changing climate on air temperature and precipitation are considered. Atmospheric deposition is also assumed to be constant in time.
- 3. Worst Case scenario. In this scenario, a socio-economic impact factor is multiplied to the future nutrient loads calculated with E-HYPE under the RCP 4.5 and RCP 8.5 scenarios (2006-2098). Socio-economic impact factors summarize the impact from each of the Shared Socio-economic Pathways (SSPs) (O'Neill et al., 2014) on current nutrient loads and atmospheric deposition, based on the downscaling of the global trends of socio-economic drivers to the Baltic Sea region (Marianne Zandersen, pers. comm.). To represent the worst conditions, the impact factor from the so-called SSP5 was selected, representing the changes caused by a 'fossil-fuelled development' scenario. Following the assumptions of the global SSP, changes in nitrogen and phosphorus loads were calculated from the regional assumptions e.g. on population growth, changes in agricultural practices such as land and fertilizer use, expansion of sewage water treatment plants...

[revised manuscript text omitted]

---

## Referee Comment (RC2) · Anonymous Referee #2 · 5 Jun 2018

This manuscript investigated the uncertainties/changes in future projections of ecosystem processes, such as primary production, nitrogen fixation and hypoxic areas, over the Baltic Sea. The authors employed a regional ocean model with capabilities of simulating coastal and ocean biogeochemical processes, driven by output from regional coupled atmosphere-ocean climate and hydrological models. The regional models, in turn, were forced by boundary conditions from global GCMs (IPCC models). The authors argued that uncertainties in ecosystem processes originate mainly from various scenarios of nutrient load, rather than model deficiencies or future greenhouse gas emissions.

This study could be of interest to ESD readers and contribute to understandings of the uncertainties in future projections of Baltic Sea ecosystem. However, I feel that the authors' manuscript needs to be improved substantially, both in terms of their analysis and general writing on their results, before it can be published in ESD. Please see my detailed comments bellow.

Major comments: 1. So far, description of experimental configuration (section Methods in the authors manuscript) is not very clear to me. It would be better, if the authors could make a schematic diagram to illustrate how their experiments are setup. For example, they can show how the "Baltic Sea model" is forced by variables from reginal hydrological and climate models, and how the regional models are forced by global GCMs. A good schematic diagram could help readers tremendously.

2. I suggest the authors also validate their regional ocean experiments individually against historical observations of ocean temperature, salinity, sea-ice cover, etc. Currently, it is done as ensemble mean and standard deviation compared with observations (e.g., Figures in Appendix). It is beneficial to show, out of the four GCMs, which provides a better forcing fields for the regional model during historical period? How do the biases in GCMs propagate to the regional ocean model used by the authors?

3. Sea-ice processes were not mentioned at all in the current manuscript. In fact, sea ice plays an important role in the budget of heat, freshwater, carbon and nutrients over the Baltic Sea (Granskog et al., 2006; Vihma and Haapala, 2009). I think the authors should discuss how sea ice is treated in their experimental setup, how well sea-ice processes are simulated in their model, and how response in sea ice influences their results.

4. The authors keep using the word "model deficiencies" when discussing results from experiments forced by four GCMs but fail to describe what exactly these model deficiencies are, and how these deficiencies influence regional simulation of the physical climate and biogeochemistry over the Baltic Sea. Also, spread between multiple mod-
els is not always the same as deficiencies in models. Internal variability could also contribute to some of the multiple-model spread. Model deficiency are usually discussed with some exact physical/biogeochemical processes.

5. The authors simply described results from their experiments and did not provide in-depth analysis/assessment on physical and biogeochemical processes producing these results. Some degree of mechanistic interpretation of their results could be interesting.

Minor comments: The writing of the current manuscript needs improvement. I do have some editing suggestions, but I feel there is no point in addressing them in this early stage.

Reference: Granskog, M., Kaartokallio, H., Kuosa, H., Thomas, D. N., & Vainio, J. (2006). Sea ice in the Baltic Sea–a review. Estuarine, Coastal and Shelf Science, 70(1-2), 145-160. Vihma, T., & Haapala, J. (2009). Geophysics of sea ice in the Baltic Sea: A review. Progress in Oceanography, 80(3-4), 129-148.

---

## Author Comment (AC2) · 29 Jun 2018

**Answers to the comments submitted by anonymous Referee #2 concerning the manuscript entitled "Uncertainties in projections of the Baltic Sea ecosystem driven by an ensemble of global climate models" by Sofia Saraiva et al.**

**Anonymous Referee #2

**We acknowledge the comments of the reviewer. We rephrased large parts of the text and added a figure with a conceptual diagram of the modelling approach, as**

[Figure]

**the reviewer suggested. We hope that the manuscript is now better readable for the reviewer and the readership of ESD. The answer to specific comments are addressed below**.

*This manuscript investigated the uncertainties/changes in future projections of ecosystem processes, such as primary production, nitrogen fixation and hypoxic areas, over the Baltic Sea. The authors employed a regional ocean model with capabilities of simulating coastal and ocean biogeochemical processes, driven by output from regional coupled atmosphere-ocean climate and hydrological models. The regional models, in turn, were forced by boundary conditions from global GCMs (IPCC models). The authors argued that uncertainties in ecosystem processes originate mainly from various scenarios of nutrient load, rather than model deficiencies or future greenhouse gas emissions. This study could be of interest to ESD readers and contribute to understandings of the uncertainties in future projections of Baltic Sea ecosystem. However, I feel that the authors' manuscript needs to be improved substantially, both in terms of their analysis and general writing on their results, before it can be published in ESD. Please see my detailed comments bellow. Major comments:*

1. *So far, description of experimental configuration (section Methods in the authors manuscript) is not very clear to me. It would be better, if the authors could make a schematic diagram to illustrate how their experiments are setup. For example, they can show how the "Baltic Sea model" is forced by variables from reginal hydrological and climate models, and how the regional models are forced by global GCMs. A good schematic diagram could help readers tremendously.*
**Answer: A new figure was added to the manuscript, illustrating the hierarchy of models used in this study.**
1. *I suggest the authors also validate their regional ocean experiments individu-*

*ally against historical observations of ocean temperature, salinity, sea-ice cover, etc. Currently, it is done as ensemble mean and standard deviation compared with observations (e.g., Figures in Appendix). It is beneficial to show, out of the four GCMs, which provides a better forcing fields for the regional model during historical period? How do the biases in GCMs propagate to the regional ocean model used by the authors?*

**Answer: The authors agree with the reviewer on the importance of comparing individual performances obtained by the use of the different CGMs downscales. However, the topic would, per se, deserve a separate manuscript. After a selection of available models that could reasonably reproduce the historical period, the main goal of the present study was to compare the uncertainty induced by different nutrient scenarios with the inherent uncertainty induced by the use of different GCMs. For that reason and to enhance the comparison between scenarios in the future simulation rather than the validation of the different models, the paper gives more focus on the ensemble than on the individual performances. However, although not explored thoroughly, the individual results are shown for salinity and runoff (Fig. 9), hypoxic area (Fig.10) and also the GCMs individual performances in terms of average profiles of the main properties through comparison with average observations, in the supplementary material (Fig. S1). In addition, in some parts of the manuscript the reader is invited to search for model information through the reference to other studies. The main goal of these figures and the manuscript is to present the range of possible solutions, rather than the selection of the best model. In fact, as far as we can say from our results during the historical period, either the comparison of atmosphere conditions imposed in the coupled physical-biogeochemical model or its water conditions impacts, there is no unique best model to use. Each model has its own strengths and weaknesses (being better or worse to simulate particular properties or**

**features) and for that reason, we suggest that the best methodology, in this type of studies, is exactly to not select, and to use several models that could represent a range of possibilities.**

2. *Sea-ice processes were not mentioned at all in the current manuscript. In fact, sea ice plays an important role in the budget of heat, freshwater, carbon and nutrients over the Baltic Sea (Granskog et al., 2006; Vihma and Haapala, 2009). I think the authors should discuss how sea ice is treated in their experimental setup, how well sea-ice processes are simulated in their model, and how response in sea ice influences their results.*

**Answer: We agree with the reviewer on the fact that sea ice plays an important role in the budget of heat. However, in our manuscript we focus on biogeochemical cycles and eutrophication which is an important pressure in the Baltic proper where in historical climate on average the extent of sea ice cover is small. Changes were made in the text and the manuscript points now to the study from Eilola et al. (2013) where the impact of future sea ice retreat on the Baltic Sea biogeochemistry at the end of the 21st century is more thoroughly studied. Eilola et al. (2013) found an earlier onset of the spring bloom, increased wind and wave-induced resuspension and increased winter mixing in areas having reduced ice cover. Our results corroborate those findings.**

3. *The authors keep using the word "model deficiencies" when discussing results from experiments forced by four GCMs but fail to describe what exactly these model deficiencies are, and how these deficiencies influence regional simulation of the physical climate and biogeochemistry over the Baltic Sea. Also, spread between multiple models is not always the same as deficiencies in models. Internal variability could also contribute to some of the multiple-model spread. Model deficiency are usually discussed with some exact physical/biogeochemical processes.*

**Answer: We agree with the reviewer that the concept of model deficien-**

cies is not defined properly in the submitted version of the manuscript. Changes were made in the introduction of the manuscript to clarify that concept: GCM deficiencies are considered in this study as the uncertainties inherited to the Regional Climate Model, and consequently to the coupled physical-biogeochemical model, from the GCMs projections, that are used as boundary forcing. We added: These deficiencies are defined as model shortcomings that affect in the dynamical downscaling approach the performance of the RCMs and consequently of the coupled physical-biogeochemical ocean model. However, GCMs are biased not only due to model deficiencies but additionally such models cannot be considered to be in phase with the real climate due to decadal climate variations (Deser etal. 2014). Together with this natural variability uncertainties in GCM initialization will shift the period of the GCM climate to a different state. Moreover, even though there are internationally coordinated protocols (e.g. CMIP) for how to equilibrate GCMs, the tuning strategy of a GCM varies widely as well as the used observational reference data sets do (Hourdin etal. 2017, Schmidt etal. 2017). Likewise, large scale patterns reflecting key climate characteristics are the tuning target rather than the fit with the region of interest of the RCM (Mauritsen etal. 2012). The above mentioned deficiencies sum up and translate into RCM deficiencies via the boundary forcing. We investigate in our study only the combined impacts on scenario simulations of Baltic Sea biogeochemical cycles.

4. *The authors simply described results from their experiments and did not provide in-depth analysis/assessment on physical and biogeochemical processes producing these results. Some degree of mechanistic interpretation of their results could be interesting.*

**Answer: Since the Baltic Sea has been intensively studied in the last years, literature is vast on the description of the main processes influencing the dynamics of the ecosystem and the authors consider that there is no need**

to describe those in detail. However, we agree with the reviewer that the manuscript did not explain in detail the mechanistic reasons behind the results. Changes were done in the text to improve the understanding of the underlying processes behind the results, particularly on the subsection on the biogeochemical variables under the Results section.

5. *The writing of the current manuscript needs improvement. I do have some editing suggestions, but I feel there is no point in addressing them in this early stage.*

**Answer: We rephrased large parts of the text and we hope that the manuscript is now better structured and easier to read following the suggestions of both reviewers.**